# OpenReview forum: "Direct Judgement Preference Optimization"
_ICLR.cc/2025/Conference — Submitted to ICLR 2025_

### Official Review · Reviewer_Ay6r · 2024-10-17

**Soundness:** 3
**Presentation:** 3
**Contribution:** 3
**Rating:** 6
**Confidence:** 5

**Summary:**

The paper explores the use of preference learning algorithms (DPO) to enhance the evaluation/critique capabilities of LLMs. By relying on human-annotated judgment scores to automatically assess the quality of model-generated critiques, they are classified into chosen or rejected categories, facilitating the preference learning. Additionally, the authors designed three training methods: Chain of Thought (CoT), standard judgment, and response deduction to optimize the model, achieving impressive experimental results on 13 benchmarks, covering pairwise, point-wise, and binary classification evaluation protocols.

**Strengths:**

* This paper thoroughly analyzes the contribution of three different training examples to enhancing model evaluation and reflection capabilities: CoT, standard judgment, and response deduction.
* The authors conducted extensive testing on 13 benchmarks covering three types of evaluation protocols: pairwise, pointwise, and classification, with experimental results demonstrating superiority over existing baselines.
* The proposed method maintains good reliability in terms of bias in LLM-as-a-judge.

**Weaknesses:**

### 1. Some Prior Works Need to Be Discussed

Using preference learning for optimization sounds impressive, but some prior work have already utilized preference learning to enhance the evaluation/critique capabilities of LLMs, such as CriticGPT  [1] and Themis [2].

1. CriticGPT employs a standard RLHF workflow to improve the evaluation/critique capabilities of LLM in the code generation task, relying heavily on cost human annotations.

2. Similarly, Themis also utilizes DPO to enhance the evaluation capabilities of language models. Themis also automatically construct the preference data by checking if the model's judgment scores are consistent to human-annotated judgment scores. In terms of motivation and implementation details, there is not much difference between this paper and Themis. Of course, the authors introduce three different preference examples to enhance the capabilities of LLMs.

However, from my perspective, these two works need to be carefully compared and analyzed in this paper. Although the authors can compare the Themis with proposed method (CriticGPT is a closed-sourced model), I do not recommend doing so because I believe there may be numerous variables that could affect the experimental results, such as the size of the dataset and the optimization techniques (Themis re-designs the DPO loss).

Beyond above comments, this paper, Themis, and CriticGPT heavily rely on human-annotated judgment labels to automatically construct preference datasets. Therefore, the scalability of these methods is limited. I hope to receive some detailed discussion and analysis from the authors on this aspect.



### 2. Flexibility of Evaluation Critiera

Similar to models in the Prometheus family, the current model's input includes an additional evaluation protocol except for the user query and the evaluated response. Therefore, during real-world evaluation setting, additional craft of evaluation criteria is required, as demonstrated in lines 328-329. Although Section 5.4 demonstrates that the impact of the crafted evaluation protocol on evaluation is limited, both this paper and Prometheus-like models require the input of an evaluation protocol (no matter the human-annotated or automatically generated by GPT-4). Therefore, when performing evaluation in real-world scenarios, this severely limits the scalability of the method because it may require additional supervision of humans and advanced models (From my perspective).

### 3. Coarse-grained Evaluation Aspects

As demonstrated in Section 4.2 (line 231 tp line 234), the evaluation aspects for building training dataare general quality, like facuality, helpfulness, safety, etc, which is limited and coarse-grained, compared to previous works like Auto-J and Prometheus.
For example, Auto-J introduces 58 tasks and corresponding fine-grained evaluation criteria. Prometheus generate the score-rubrics-like customized evaluation criteria for specific user queries.

### 4. Without the evaluation of  textual critques quality (Chain-of-Thought)
This paper only evaluate the quality of judgment scores, while neglect the evaluation of textual critique quality (the quality of chain-of-thought). Previous works have demonstrated that even if the judgment score is consistent with human-annotated, their chain-of-thought may contains several flaws [3]. This will introduce numerous bias and drawbacks for application of the fine-tuned critique model in some important reseach fields, like the self-improvement of LLMs.

---


[1] LLM Critics Help Catch LLM Bugs: https://arxiv.org/abs/2407.00215

[2] Themis: A Reference-free NLG Evaluation Language Model with Flexibility and Interpretability: https://arxiv.org/abs/2406.18365

[3] The Critique of Critique: https://arxiv.org/abs/2401.04518

**Questions:**

1. This paper, Themis, and CriticGPT heavily rely on human-annotated judgment labels to automatically construct preference datasets. Therefore, the scalability of these methods is limited.

2.  When performing evaluation in real-world scenarios, requiring the input of evaluation criteria limits the scalability of the method because it may require additional supervision of humans and advanced models.

I hope to receive some detailed discussion and analysis from the authors on these aspects.

---

> ### Author Response · Authors · 2024-11-18
> **Response to Reviewer Ay6r (part 1)**
>
> > Using preference learning for optimization sounds impressive, but some prior work have already utilized preference learning to enhance the evaluation/critique capabilities of LLMs, such as CriticGPT [1] and Themis [2].
>
> > Similarly, Themis also utilizes DPO to enhance the evaluation capabilities of language models. Themis also automatically construct the preference data by checking if the model's judgment scores are consistent to human-annotated judgment scores. In terms of motivation and implementation details, there is not much difference between this paper and Themis. Of course, the authors introduce three different preference examples to enhance the capabilities of LLMs.
>
> We thank the reviewer for bringing these two papers to our attention. We have updated the introduction to mention these papers and the related work section with a longer discussion, focused largely on Themis.
>
> The core difference between Themis and our work is that Themis, as the reviewer mentions, makes single-rating specific changes to the DPO loss. In particular, they introduce a margin term into the DPO loss that is based on the difference in value of the ratings between the positive sample and negative sample, modeling the probability of preferring response A or B as
>
> $$P(A \succ B) = \sigma\left(r(x, y_A) - r(x, y_B) - \alpha|R(y_A) - R(y_B)|\right).$$
>
> Above, $r$ is the parameterized reward model, $\sigma$ is the sigmoid function, and $R(y_i)$ denotes the absolute rating score contained in response $y_i$. As a concrete example, if $y_i$ = “The response is incomplete, but brings up true points. Score: 3”, then $R(y_i) = 3$.
>
> This modification of the DPO loss is specifically designed for learning from single rating evaluations: Because the outputs of judgements are numerical scores, the difference naturally captures some measure of difficulty which can be utilized as a margin. However, it is less clear how to utilize such a DPO loss in training a judge capable of other common types of evaluation settings without numerical score outputs, such as the ones we consider in this paper (pairwise, classification). An additional minor point that arises with this modified DPO loss is the introduction of a new hyperparameter, $\alpha$, which must be set in addition to the typical DPO $\beta$ during training. The effects of $\alpha$ are not characterized by the Themis authors, so we have no insight into the sensitivity of performance with respect to $\alpha$.
>
> In contrast, we propose three different forms of pairwise preference data, without making modifications to the DPO loss, as the reviewer has noted. This design was intentional: Focusing on new ways to construct the pairwise preference data for DPO allowed us to (1) train a judge capable of all three evaluation tasks simultaneously and (2) add new judgment capabilities to the judge simply by adding new types of pairwise preference data and re-training, rather than modifying loss terms. However, we recognize the similarities, and have updated our paper introduction (lines 60-63) and related work section (lines 515 - 530) accordingly.

---

> > ### Author Response · Authors · 2024-11-18
> > **Response to Reviewer Ay6r (part 2)**
> >
> > > CriticGPT employs a standard RLHF workflow to improve the evaluation/critique capabilities of LLM in the code generation task, relying heavily on cost human annotations.
> >
> > The core difference between our model and CriticGPT is that our models are explicitly trained to provide judgements, whereas CriticGPT is trained only to provide critiques. While critiques and judgements are closely related, we see a core difference between the two:
> > - Judgements provide an easily interpretable measure of response quality, whether in an absolute (single-rating) or relative (pairwise) manner.
> > - Critiques, on the other hand, discuss how model responses can be improved, but still require downstream interpretation to arrive at a notion of overall response quality.
> >
> > While this difference may seem small, it has effects on the practicality for downstream applications beyond automatic evaluation. Consider the case where multiple responses are sampled from the same model for a given prompt, and the goal is to rank such responses. This problem arises in multiple settings, such as forming preference pairs for preference optimization (akin to the experiment we conduct in Sec. 5.5) or for re-ranking model outputs at inference time. With a judge model, such ranking can be automated via relying on the judge’s score for each output in the single-rating setting, or done via a round-robin tournament in the pairwise setting, as is done with FLAMe [1] for code re-ranking (See Sec. 6.2 of the FLAMe paper).
> >
> > CriticGPT conducts a similar experiment with chat data, showing it can generalize beyond code data. However their experiments still relied on human evaluators to manually review the critiques and then re-score/score model outputs. This process is cumbersome and unscalable, highlighting the fundamental limitations of critique-only models. It would be extremely interesting to see if CriticGPT can generalize to providing judgements in a zero/few-shot manner, but unfortunately the closed-source nature of the model makes such assessment not feasible.
> >
> > [1] Foundational Autoraters: Taming Large Language Models for Better Automatic Evaluation, https://arxiv.org/html/2407.10817v1

---

> ### Author Response · Authors · 2024-11-18
> **Response to Reviewer Ay6r (part 3)**
>
> > However, from my perspective, these two works need to be carefully compared and analyzed in this paper. Although the authors can compare the Themis with proposed method (CriticGPT is a closed-sourced model), I do not recommend doing so because I believe there may be numerous variables that could affect the experimental results, such as the size of the dataset and the optimization techniques (Themis re-designs the DPO loss).
>
> In terms of experimental comparison against Themis, we believe there are two perspectives in experimental comparison: controlled-setting comparison and practitioner comparison. The first perspective is analysis conducted to determine which training recipe, for fixed preference dataset and teacher model, leads to the best downstream judgment performance. We consider this an exciting line of open research that will inform how the next generation of judge models are trained. However, prior judge models (i.e., all evaluated baselines in our paper) are not trained on a standardized preference mix. Recent efforts have produced high-performant preference datasets, such as the Skywork Reward Preference 80K dataset [2] for the pairwise evaluation setting. Such datasets may serve as a foundation for future research in controlled analysis. However, these datasets were released well after our model development was complete, and as such, we leave a standardized analysis open as future work.
>
> The second perspective is one that we (and other judge papers) implicitly take in our benchmarking efforts. Practitioners who simply want to run a model “out-of-the-box” to get evaluation results or use judge outputs for downstream alignment may heavily prioritize judge performance on standardized benchmarks over training details or dataset mix and size. In order to characterize model performance for this perspective, we run Themis on our single rating evaluation tasks and report the performance below. *We agree with the reviewer that different variables almost assuredly impact performance; See above discussion about future lines of analysis for judge training.* As such, our stated results are meant to inform those who prioritize “out-of-the-box” performance.
>
> |               | BiGGen Bench  | BiGGen Bench  | FLASK         | FLASK         | MT-Bench      | FeedbackBench | Average |
> |---------------|---------------|---------------|---------------|---------------|---------------|---------------|---------|
> |               | Human Pearson | GPT-4 Pearson | Human Pearson | GPT-4 Pearson | GPT-4 Pearson | GPT-4 Pearson |         |
> | Themis-8B     | 0.58          | 0.69          | 0.54          | 0.58          | 0.57          | 0.76          | 0.62    |
> | Our model 70B | 0.65          | 0.81          | 0.66          | 0.74          | 0.77          | 0.93          | 0.76    |
> | Our model 12B | 0.57          | 0.74          | 0.59          | 0.66         | 0.72          | 0.93          | 0.70    |
> | Our model 8B  | 0.59          | 0.71          | 0.52          | 0.60          | 0.71          | 0.92          | 0.68    |
>
>
> Overall, Themis-8B outperforms many competitive judges in single rating tasks, matching the performance of much larger models like Prometheus-2-8x7B and performing comparably to our 8B model in a subset of benchmarks.
>
> [2] Skywork-Reward-Preference dataset: https://huggingface.co/datasets/Skywork/Skywork-Reward-Preference-80K-v0.2

---

> > ### Comment · Reviewer_Ay6r · 2024-11-21
> >
> > Thank you very much for the author's response, and I greatly appreciate the additional experiments conducted by the author to address my questions. However, I have a minor concern regarding the experimental results. The proposed 8B model seems to show significantly higher correlation with GPT-4 compared to Themis-8B, but the correlation with human evaluation shows little difference or is slightly worse. Could you provide some extended explanation regarding this issue?

---

> ### Author Response · Authors · 2024-11-18
> **Response to Reviewer Ay6r (part 4)**
>
> > Beyond above comments, this paper, Themis, and CriticGPT heavily rely on human-annotated judgment labels to automatically construct preference datasets. Therefore, the scalability of these methods is limited. I hope to receive some detailed discussion and analysis from the authors on this aspect.
>
> In terms of limitations, we agree with the reviewer that the dependence on human labels limits scalability in the field of auto-evaluation in general, including our work and others mentioned. There are many exciting lines of future work that we wish to explore to overcome the limitations of human labels, especially as the pace of LLM development accelerates. Concurrent work (Self-taught-evaluator [3]) provides an initial exploration into self-taught evaluators that eschew the need for human annotations. However, the empirical performance of the pairwise-specific Self-taught-evaluator still lags our multi-faceted models in pairwise comparisons, indicating a gap in purely synthetic training approaches. Below, we sketch out an approach to re-purpose existing judge models to create the training samples for the next generation of auto-evaluators.
>
> An interesting line of future work in auto-evaluation could be to utilize all judge models as a “crowd” to conduct crowdsourcing. That is, to construct new pairwise preference datasets, we evaluate responses with existing judge model checkpoints (whether pairwise or rating), then aggregate responses. This approach diversifies the source of automated labels appealing to the wisdom of a model crowd, rather than relying on a singular model like GPT-4o to generate labels. Because each crowd member is a model, new exciting approaches in aggregating responses are enabled. Rather than simple majority vote, we can weight each model response by confidence heuristics, like model uncertainty (measured via logits), model size (e.g., give more weight to a 70B model vs. an 8B model), or even model performance on benchmarks (e.g., perhaps we trust a model that achieves 90 on RewardBench more than a model that achieves 75). This entire process can be viewed as a crowd-based self-improvement process, with previous generations of judge models serving as previous “iterations” in self-improvement.
>
> The added bonus is that every new judge model checkpoint released is another member of the crowd, encouraging researchers and practitioners to collectively work towards building gold-standard auto-evaluators. We hope that our models can help overcome current limitations of training judge models. Towards this end, we are planning on releasing our model checkpoints for research purposes at a later date, once we have obtained proper institutional approval, which is currently pending.
>
> [3] Self-Taught Evaluators, https://arxiv.org/abs/2408.02666
>
> ---
> > Similar to models in the Prometheus family, the current model's input includes an additional evaluation protocol except for the user query and the evaluated response. Therefore, during real-world evaluation setting, additional craft of evaluation criteria is required, as demonstrated in lines 328-329. Although Section 5.4 demonstrates that the impact of the crafted evaluation protocol on evaluation is limited, both this paper and Prometheus-like models require the input of an evaluation protocol (no matter the human-annotated or automatically generated by GPT-4). Therefore, when performing evaluation in real-world scenarios, this severely limits the scalability of the method because it may require additional supervision of humans and advanced models (From my perspective).
>
> We thank the reviewer for bringing up an interesting point. In our view, applying judge models for automatic evaluation requires humans to first define what criteria to use in the evaluation, and to embed such criteria in the prompt to the judge model. We define this characterization of criteria as the evaluation protocol, and is universal in evaluating model responses, regardless of if judge models or human evaluators are used.
>
> We emphasize that unlike Prometheus, evaluation rubrics or reference answers are not required for our model. However, if rubrics or reference answers are available, they can be incorporated in evaluation. In our setting, the evaluation protocol can simply be natural language explanation of what to emphasize (e.g., “focus on how well the output satisfies the user’s instructions”) or a specific manner to conduct evaluation (e.g., PRePair-style prompting of enumerating pros and cons independently before producing a pairwise comparison response). We have updated our paper to explicitly clarify how such protocols differ from Prometheus in lines 142 - 145.
>
> In light of this, we consider the fact that our judge model exhibits relatively stable performance across evaluation protocols in Sec. 5.4 as a strong indicator of the flexibility of our judge model to accommodate evaluation criteria, whether that be task-specific or a fixed evaluation strategy (e.g., PRePair).

---

> > ### Author Response · Authors · 2024-11-18
> > **Response to Reviewer Ay6r (part 5)**
> >
> > > As demonstrated in Section 4.2 (line 231 tp line 234), the evaluation aspects for building training data are general quality, like facuality, helpfulness, safety, etc, which is limited and coarse-grained, compared to previous works like Auto-J and Prometheus. For example, Auto-J introduces 58 tasks and corresponding fine-grained evaluation criteria. Prometheus generate the score-rubrics-like customized evaluation criteria for specific user queries.
> >
> > We thank the reviewer for raising this point, as this is an instance where our writing was not sufficiently clear. For each training dataset, we have preserved the original level of evaluation granularity, whether that be coarse-grained evaluation or very fine-grained evaluation. We briefly mention that original annotation instructions are preserved in Section 2 (lines 139 - 140), but agree that it would be best to explicitly state in this section as well For datasets like Prometheus, we preserve original rubrics and reference answers. We additionally preserve all evaluation attributes for datasets with multiple attributes, like HelpSteer2. That is, our judges are trained at varying levels of evaluation coarseness, allowing them to generalize across evaluation coarseness AND domains. We have added a sentence explicitly stating this preservation of evaluation coarseness (line 227).
> >
> > ---
> >
> > > This paper only evaluate the quality of judgment scores, while neglect the evaluation of textual critique quality (the quality of chain-of-thought). Previous works have demonstrated that even if the judgment score is consistent with human-annotated, their chain-of-thought may contains several flaws [3]. This will introduce numerous bias and drawbacks for application of the fine-tuned critique model in some important reseach fields, like the self-improvement of LLMs.
> >
> >
> > This is a valuable point raised by the reviewer! We conducted an additional experiment using the referenced MetaCritique framework to evaluate the quality of the critiques produced by our models, and have updated our paper to add this experiment. MetaCritique is focused on evaluating if model-generated answers to user questions are correct or not. As such, we prompt our models to conduct classification evaluation, where we present the judge with the Q&A pair and ask the model to produce a critique and a binary yes/no label for correctness.
> >
> > We additionally evaluate Self-taught-evaluator-70B and Themis. For Self-taught-evaluator, we prompt the judge to perform the same binary classification task as our judge models. For Themis, we prompt the judge to perform single-rating evaluation (rate the response based on the user’s question) and classification, and report both results. While the classification approach is more natural for this setting, Themis was trained exclusively to perform single rating evaluation, and as such, we experiment with both.
> >
> > We report performance below, using reported numbers from the MetaCritique leaderboard for other baselines. We also have updated our paper with these results and extended discussion in Appendix C.7.
> >
> > |                           | Meta-Precision | Meta-Recall | Meta-F1 Score |
> > |---------------------------|----------------|-------------|---------------|
> > | Auto-J (previous best)    | 76.43          | 70.65       | 71.14         |
> > | Shepherd human critiques  | 83.19          | 60.65       | 64.02         |
> > | Themis-8B-Rating          | 77.98          | 53.31       | 58.83         |
> > | Themis-8B-Classification  | 76.54          | 55.05       | 60.48         |
> > | Self-taught-evaluator-70B | 77.60          | 59.60       | 62.99         |
> > | Our model 70B             | 93.10          | 70.54       | 77.60         |
> > | Our model 12B             | 89.15          | 68.86       | 74.04         |
> > | Our model 8B              | 83.04          | 64.46       | 69.52         |
> >
> >
> > The Meta-Precision metric is meant to measure the factuality of the critique produced by the judge model, whereas the Meta-Recall metric is meant to measure how well your critique matches one produced by GPT-4 in terms of substance and correctness. The Meta-F1 score aggregates Meta-Precision and Meta-Recall into one measure of critique quality. Overall, our three models exhibit strong performance, with our 12B and 70B models producing more factual critiques (Meta-Precision) than the previous best models and all three of our models outperforming human critiques from source datasets. Auto-J achieves the highest Meta-Recall score, which is likely expected since (1) Auto-J is trained on critiques generated from GPT-4, and (2) Meta-Recall measures similarity of generated critiques to those generated by GPT-4.
> >
> > Notably, the fact that our judge models produce far more factual critiques than comparable baselines indicates that using such critiques for downstream model development or self-improvement will lead to less bias/incorrect reasoning than other judge models.

---

> > > ### Comment · Reviewer_Ay6r · 2024-11-21
> > >
> > > Thank you very much for the additional experiments, which have resolved my concerns about these questions.

---

> > ### Comment · Reviewer_Ay6r · 2024-11-21
> >
> > Thank you for your impressive response, which has resolved my concerns about these issues.

---

> ### Author Response · Authors · 2024-11-21
> **Follow-up response to Reviewer Ay6r**
>
> >Thank you very much for the author's response, and I greatly appreciate the additional experiments conducted by the author to address my questions. However, I have a minor concern regarding the experimental results. The proposed 8B model seems to show significantly higher correlation with GPT-4 compared to Themis-8B, but the correlation with human evaluation shows little difference or is slightly worse. Could you provide some extended explanation regarding this issue?
>
>
> We are happy to provide additional explanations. We begin by noting that our models and Themis are both trained on a mix of human-annotated and model-annotated single rating data. However, Themis only used GPT-4 data that closely matched human annotations. From the Cross-Validation paragraph in Section 3.2 of the [Themis paper](https://arxiv.org/pdf/2406.18365):
>
>  >We prioritized samples where the two evaluation ratings were close, which reflected the high consistency between evaluations from humans and GPT-4 and also indicated the potential strong reliability of the samples.
>
> Therefore, it is likely that Themis discarded many samples labeled by GPT-4 that exhibited disagreement with human annotations, skewing the distribution of GPT-4 annotated training data to be close to human annotations. This choice likely explains why Themis exhibits strong human correlation but relatively weaker GPT-4 correlation: Samples that would allow Themis to potentially bridge the gap in correlation with GPT-4 are not used in training.
>
> The fact that our 8B model roughly matches the performance of Themis-8B in terms of human-annotation agreement (+0.01 in BiGGen Bench, -0.02 in FLASK relative to Themis) *demonstrates that the additional cross-validation step of removing GPT-4 samples that do not agree with human annotations is not necessary*: Judges can generalize their decision making to both types of annotators. This generalization to correlate with GPT-4 annotations may be desirable, as we describe next.
>
> Here, we provide our rationale for including GPT-4 annotated benchmarks in our evaluation suite. While agreement with human annotations is the ultimate goal of automatic evaluation, measuring agreement with GPT-4 annotations is also a scenario that has practical importance. Consider a practitioner that currently uses GPT-4 as a judge model, but is looking to move to a smaller, self-hosted judge model for practical reasons (cost, inference time, ability to continuously fine-tune, removing reliance on an external API, removing dependence on an opaque model-backend which may change with short notice, etc). If the practitioner’s current system works sufficiently well with GPT-4 as a judge, they may want a drop-in replacement that behaves like GPT-4 in its assessments. As a result, the practitioner may prioritize higher correlation with GPT-4 outputs. In this scenario, measurement with human annotators is **implicit** in that the practitioner is happy with using GPT-4 as a judge, implying that GPT-4’s ratings correlate well with human ratings.
>
> We believe the above rationale is an important point to emphasize, and will update Appendix A (Evaluation dataset overview) in our paper with a discussion of this importance.
>
> Lastly, training details are important in this discussion. As we’ve highlighted, Themis-8B is (1) trained exclusively single-rating tasks and (2) trained using single-rating specific modifications to the DPO training loss. This training approach makes Themis a very competitive single-rating model, especially compared to other 8B models. Our models are less specialized in that they are trained for other tasks and do not make single-rating specific modifications to the training loss. Despite less specialization, our models are competitive with Themis, with the 8B model performing roughly on par with Themis on tasks (human agreement on single-rating tasks) that Themis was trained to excel at.
>
> If this discussion and our earlier answers have addressed your concerns, we would greatly appreciate it if you would consider raising your score. We are also happy to answer any additional questions!

---

> > ### Comment · Reviewer_Ay6r · 2024-11-22
> >
> > Thank you for the clarification. My concern has been resolved to some extent.

---

### Official Review · Reviewer_azFf · 2024-11-03

**Soundness:** 4
**Presentation:** 4
**Contribution:** 4
**Rating:** 8
**Confidence:** 4

**Summary:**

The paper explores enhancing large language models (LLMs) as generative judges for evaluating model responses and generating critiques. It introduces direct preference optimization (DPO) using diverse human and synthetic judgments to improve evaluation across pairwise, single ratings, and binary classification tasks. The trained models, especially the 8B parameter model, outperform strong baselines and specialized judge models, showing robust performance and adaptability across various benchmarks.

**Strengths:**

1. The problem of automated evaluation is important and well studied in recent years. Thus, the work is well motivated.
2. Use of DPO using positive and negative pairs to improve LLM eval is novel. Response deduction is also a nice idea.
3. The framework is comprehensive -- works on 3 eval setttings -- Single Rating, Pairwise Comparison, and Classification.
4. Comprehensive eval on seven pairwise comparison benchmarks, four single rating evaluation benchmarks, and two classification benchmarks. The results are convincingly better compared to our SOTA models and even gpt4o.
5. Bias evaluation results, ablations, flexible prompting strategies, downstream model improvements -- are all nice additions in the paper.

**Weaknesses:**

1. It will be interesting to see how easy vs hard negatives can help. Given the current way of generating negatives, it is unclear if they were hard enough and if a mix of easier and hard negatives could help improve further.
2. Choice of 70%:15%:15% for DCoT, DStd and DDed sounds somewhat arbitrary. Can you provide further reasoning behind this?
3. Broadly a comparison to evaluation systems with multiple models is not expected as such, but still it will be nice to compare and check how does your method works compared to methods like reconcile.

Typos:
1. citep vs citet should be used properly. E.g. line 41 should have "GPT-4 (OpenAI, 2023)" and not "GPT-4 OpenAI (2023)".
2. Notations M_t vs M_{teacher} or M'_{teacher} do not match between Section 3 and Fig. 2. Same for student model.

**Questions:**

1. Is it possible to quantify hardness of negatives and does that correlate with evaluation accuracy?
2. Choice of 70%:15%:15% for DCoT, DStd and DDed sounds somewhat arbitrary. Can you provide further reasoning behind this?

---

> ### Author Response · Authors · 2024-11-18
> **Response to Reviewer azFf (Part 1)**
>
> > It will be interesting to see how easy vs hard negatives can help. Given the current way of generating negatives, it is unclear if they were hard enough and if a mix of easier and hard negatives could help improve further.
>
> Thank you for the interesting question! This is something that we explored at an earlier stage in our model development, and have updated our paper to include this result in Appendix C.6. In particular, we trained two 8B judge models with different teacher model setups. Judge (a) used preference data with both positive and negative pairs sampled from the same teacher model (Llama-3.1-70B). Judge (b) used preference data where positive samples are from Llama-3.1-70B and negative samples are from Llama-3.1-8B. Therefore, one can consider Judge (a) to be trained on “harder” negative samples, whereas Judge (b) is trained on “easier” negative samples.
>
> |                       | Average pairwise | Average pairwise consistency | Average Pearson | Average classification |
> |-----------------------|------------------|------------------------------|-----------------|------------------------|
> | Hard preference pairs | 78.83            | 85.94                        | 0.68            | 85.48                  |
> | Easy preference pairs | 77.56 (↓ 1.27)   | 80.70 (↓ 5.24)               | 0.67 (↓ 0.1)    | 84.54 (↓ 0.94)         |
>
> We report evaluation results for these two models in the above table and have added an additional ablation section in Appendix C.6 with additional discussion. Note that these are earlier checkpoints, and as such, results for Judge (a) differ from our final reported numbers. In particular, training with a weaker teacher model for negative samples resulted in drops in performance across the board, with the most notable drop coming from pairwise comparison consistency, highlighting that harder preference pairs play a critical role in mitigating certain judge biases.
>
> We recognize that with a fixed teacher model, i.e., the Judge (a) setting above, samples may still vary in difficulty. We consider methods for identifying hard negatives in generating pairwise preference data for judges an exciting avenue of future research.
>
> ---
>
> > Choice of 70%:15%:15% for DCoT, DStd and DDed sounds somewhat arbitrary. Can you provide further reasoning behind this?
>
> Thank you for highlighting this important aspect! The choice of data distribution reflects our focus on enhancing the reasoning capabilities and interpretability of our LLM judge. Since reasoning is a crucial component for accurate and transparent evaluations, we prioritized D_CoT to ensure the model effectively learns to generate the Chain-of-Thought critique before making the final judgment. At the same time, we maintained a balance with direct supervision ($D_{Std}$) and response understanding ($D_{Ded}$), as both have been shown to significantly contribute to overall performance improvements.
>
> ---
>
> > citep vs citet should be used properly. E.g. line 41 should have "GPT-4 (OpenAI, 2023)" and not "GPT-4 OpenAI (2023)".
> Notations M_t vs M_{teacher} or M'_{teacher} do not match between Section 3 and Fig. 2. Same for student model.
>
> Thank you for pointing out these issues. We have reviewed the paper to make sure citep and citet are used properly. We have also standardized the notation for the teacher/student model in Section 3 to match those in Fig. 2.

---

> ### Author Response · Authors · 2024-11-18
> **Response to Reviewer azFf (Part 2)**
>
> > Broadly a comparison to evaluation systems with multiple models is not expected as such, but still it will be nice to compare and check how does your method works compared to methods like reconcile.
>
> Thank you for the suggestion! We have conducted an additional experiment where we compare our model against ChatEval [1], a multi-model system for making pairwise comparisons. ChatEval is similar to ReConcile, where it uses multiple powerful language models to conduct back-and-forth debate before arriving at a final answer. However, unlike ReConcile, which focuses on improving reasoning capabilities, ChatEval is specifically designed for making pairwise comparison judgements of other language model outputs. As such, we run ChatEval on our 7 pairwise comparison benchmarks.
>
> Specifically, we adopt the default setup of ChatEval, where we have 2 agents, both GPT-4, which assume different roles: a “General-public agent” and a “Critic agent”. These agents debate sequentially for four rounds. We keep the default evaluation prompts provided by ChatEval. At the end of the debate, both agents output a final score for each output. We average the scores for both agents, and set the winner as the higher-scoring response. If the final score is tied, we break ties arbitrarily by arbitrarily choosing a winner. Because of the high inference time and API cost of this evaluation, we only run each pairwise comparison benchmark once, so we are unable to report consistency values.
>
> |               | RewardBench | InstruSum | Auto-J | HHH   | LFQA  | EvalBiasBench | PreferenceBench | Average |
> |---------------|-------------|-----------|--------|-------|-------|---------------|-----------------|---------|
> | ChatEval      | 78.0            | 62.04     | 61.13       | 85.07 | 72.69 | 66.25         | 82.23       | 72.49         |
> | Our model 70B | 92.7        | 82.73     | 63.51  | 94.57 | 75.00 | 85.00         | 96.25           | 84.25   |
> | Our model 12B | 90.3        | 75.18     | 62.50  | 92.31 | 71.15 | 82.50         | 96.85           | 81.49   |
> | Our model 8B  | 88.7        | 74.94     | 60.34  | 94.12 | 68.85 | 85.00         | 94.39           | 80.91   |
>
>
> Despite being powered by a much larger model (GPT-4) and relying on multiple turns of reasoning, ChatEval still underperforms even our 8B model in aggregate, trailing by 8.42 points in terms of aggregate performance. ChatEval struggles in key domains such as safety (85.07 on HHH vs. 94.12 for our 8B model) and bias (66.25 on EvalBiasBench vs. 85.00 for our 8B model), highlighting both the impressive capabilities of our judge models and the shortcomings of prompting more capable, general-purpose LLMs. Our results are consistent with results  reported in systematic investigations into judging protocols, where multi-agent, multi-round setups often lag other simpler prompting-based approaches; see the performance of ``multi-role-round{1,2}‘’ systems in Table 9 in [2].
>
> [1] ChatEval: Towards better LLM-based evaluators through multi-agent debate, https://arxiv.org/abs/2308.07201
>
> [2] REIFE: Re-evaluating Instruction-Following Evaluation, https://arxiv.org/pdf/2410.07069

---

> > ### Comment · Reviewer_azFf · 2024-11-26
> >
> > I really appreciate the time the authors spent on doing these extra experiments. Thank you for your easy vs hard negatives experiments. Also, thanks for comparing with ChatEval.
> > I have already provided a high rating. I will retain the rating.

---

### Official Review · Reviewer_xfuc · 2024-11-05

**Soundness:** 3
**Presentation:** 1
**Contribution:** 2
**Rating:** 3
**Confidence:** 4

**Summary:**

This paper curates the preference data to train an LLM as an evaluation judge using DPO, and demonstrates the effectiveness of the trained models in different sizes (e.g., 8B, 12B, 70B) on three types of evaluation tasks (e.g., pairwise comparison, single rating and classification). The contribution lies in the method of creating preference data for three evaluation tasks using a collection of labeled data from diverse data sources (annotated by humans or other models).

**Strengths:**

**Originality v.s. Significance**: This paper proposes to process existing annotated data into the preference data format, and train an LLM as a judge using DPO on the curated data. It’s a straightforward application of DPO to the task of training an LLM for evaluation. The novelty lies in the way of constructing preference data from existing labeled datasets for three evaluation tasks — but see the novelty concerns in the weakness section below. The technical novelty is weak, compared to the significance of releasing a good evaluation model to the community. However, I don’t find the statement of releasing the model checkpoints and the curated data from the paper. It’d be better to clarify this explicitly.

**Quality**: Extensive experiments and analyses have been conducted on seven pairwise comparison benchmarks, four single rating benchmarks, and two classification benchmarks.

**Weaknesses:**

**Novelty of data curation**: As the main contribution of this paper is data curation, it’s better to clarify the difference between this work and existing studies on data curation for LLM evaluation. The key differences are still unclear even after reading the description in Lines 136-150. From Table 1, existing studies have explored the three evaluation tasks (Vu et al., 2024), and have used both SFT and DPO (Want et al, 2024c). This paper seems incremental to combine what has been done in existing studies at a larger scale of training data.

**Presentation**: It’s hard to interpret the results and the evaluation with other baseline models (see details in the question section below). The tables and figures are far away from their corresponding text descriptions, making it hard to read them at the same time.

**Questions:**

1. In Line 315, what do you mean by evaluating models trained for single ratings on classification? How do you use a model trained for single ratings to evaluate classification tasks?
2. In Line 314, “All baselines are evaluated only on the tasks they are trained for”.  Can you report which tasks these baselines are trained for? How can you get the results for the tasks that these baselines are not trained for in Table 1 & 2? Do you get the baseline models’ results from their papers or run the baseline models by yourself? Do you test your models on those benchmarks without further fine-tuning your models on the benchmark data at all?
3. In Table 5, what are the 6 types of biases? And How do you measure these biases? I understand the position bias can be measured by the consistency of choosing responses even if the order is swapped. But what about the other biases?
4. Line 464 claims that “we plot the average performance across all three evaluation tasks when removing each training task”, but Fig 4 may miss the SFT+DPO w/o CoT critique.

---

> ### Author Response · Authors · 2024-11-18
> **Response to Reviewer xfuc (Part 1)**
>
> > Novelty of data curation: As the main contribution of this paper is data curation, it’s better to clarify the difference between this work and existing studies on data curation for LLM evaluation. The key differences are still unclear even after reading the description in Lines 136-150. From Table 1, existing studies have explored the three evaluation tasks (Vu et al., 2024), and have used both SFT and DPO (Want et al, 2024c). This paper seems incremental to combine what has been done in existing studies at a larger scale of training data.
>
> > The technical novelty is weak, compared to the significance of releasing a good evaluation model to the community. However, I don’t find the statement of releasing the model checkpoints and the curated data from the paper. It’d be better to clarify this explicitly.
>
> We plan on releasing model checkpoints, once we have obtained proper institutional approval (currently pending). We have added a footnote to Page 1 that clearly states this, as well as a link to an anonymized version of our evaluation code. We apologize for this oversight! Here, we’ll briefly discuss the novelty of our work in relation to prior work:
>
> Our work is different from prior work in that we are, to our knowledge, the first work to explore utilizing DPO to train an explanation-generating judge LLM for multiple evaluation settings. Our contributions are (1) utilizing DPO for training a multi-faceted judge model capable of 3 different evaluation tasks, (2) an exploration of different DPO tasks to enhance the judging and reasoning abilities of judge models, and (3) eventual release of model checkpoints for research purposes, once we have obtained proper institutional approval, which is currently pending. We have re-written portions of the introduction to clarify; please see lines 73-75 and lines 82-87.
>
> In terms of novelty, Self-taught-evaluators (Wang et al, 2024c) is concurrent work, for which the DPO training recipe was released after our model development was completed. In fact, the [paper from Wang et al, 2024c](https://arxiv.org/abs/2408.02666) only mentions iterative SFT. However, an update to their [code-base](https://github.com/facebookresearch/RAM/tree/main/projects/self_taught_evaluator)
> on September 26 revised the training process to include DPO. Despite how close this update was to the ICLR submission deadline (October 1), we immediately evaluated this model to ensure fair comparison. Because this update was made so close to the deadline, ICLR guidelines are explicit in this characterization as concurrent work: ‘’We consider papers contemporaneous if they are published within the last four months.” See bottom of [this link](https://iclr.cc/Conferences/2025/ReviewerGuide). The caption for Table 1 has noted the ambiguity in the training method of this model.
>
> In terms of technical differences, Self-taught-evaluators use iterative DPO (according to the Github link), requiring multiple rounds of training, to train a judge model that only performs pairwise comparison evaluation. For preference data, they only use pairs of {correct critique + judgment, incorrect critique + judgment}. In other words, Self-taught-evaluators only train their model with preference pairs that include critiques. Our work, in contrast, additionally includes pairs with only standard judgments and a reverse response deduction task (Training Tasks 2 and 3 in Figure 1). The fact that our judge, which is not tailored to pairwise evaluation, outperforms this pairwise-specific model demonstrates (1) the effectiveness (and relative simplicity) of our training recipe, specifically the importance of our additional training tasks.
>
> ---
> > Presentation: It’s hard to interpret the results and the evaluation with other baseline models (see details in the question section below). The tables and figures are far away from their corresponding text descriptions, making it hard to read them at the same time.
>
> Thank you for your feedback on formatting. We have updated the paper so that tables and figures are closer to their reference points. We will continue to experiment with paper layout to ensure our final version is as readable as possible.

---

> ### Author Response · Authors · 2024-11-18
> **Response to Reviewer xfuc (Part 2)**
>
> > In Line 314, “All baselines are evaluated only on the tasks they are trained for”. Can you report which tasks these baselines are trained for?
>
> Some existing judge models are only trained for certain evaluation settings, like pairwise comparison evaluation only. We have reported different judge model capabilities for the baselines we evaluate in the second column of Table 1. We have updated the paper in Section 4.3 with an explicit reference to Table 1 for tasks, and provide an example of what we mean by “All baselines are evaluated only on the tasks they are trained for” in lines 285 - 288.
>
> ---
> > In Line 315, what do you mean by evaluating models trained for single ratings on classification? How do you use a model trained for single ratings to evaluate classification tasks?
>
> For all reported numbers, we run publicly available checkpoints in our evaluation setup. To evaluate single rating judges on classification tasks, we prompt the judge model to perform the (binary) classification task directly. That is, the model is prompted to output the natural language text “Yes” or “No”, which is then compared to the ground-truth label. Because a vast majority of judge models are not trained to do classification evaluation, we evaluate single rating judges on classification tasks via the aforementioned prompting approach. We emphasize that we are not using these judge models as logit-based classifier models. Rather, we prompt the model to output a natural language decision. We have made this explicit in our updated paper in lines 289 - 291.
>
> We choose to evaluate single rating judges in this setting because of the pointwise nature of both classification and single rating tasks: they evaluate responses in isolation, without any other answer to compare against, unlike the pairwise comparison evaluation setting. Judge models that are trained exclusively on the pairwise comparison evaluation setting may not generalize to the single rating or classification settings, as we show below
>
> ---
> > How can you get the results for the tasks that these baselines are not trained for in Table 1 & 2? Do you get the baseline models’ results from their papers or run the baseline models by yourself? Do you test your models on those benchmarks without further fine-tuning your models on the benchmark data at all?
>
> We run all baselines and our judge models ourselves under a fixed evaluation setup. We have attached a link to an anonymized version of the evaluation code in the footnote on Page 1 and have added a note to the text (line 265) to emphasize that we run each baseline model ourselves.
>
> We do not conduct additional finetuning, as judge models are typically released as models ready to use “out-of-the-box”. We ensure that we do not finetune our own judges on benchmark data; i.e., we ensure no test set data contamination. For each benchmark, we conducted a comprehensive evaluation of any data sources used to create such benchmarks, and created our training set with these data sources in mind. In particular, we take caution to only use train or dev sets, and check that such samples do not appear in benchmarks.
>
> We note that with the exception of evaluating models capable of single-rating judgements on classification tasks, we do NOT have results for tasks that baselines are not trained on. Here, we demonstrate that the choice to evaluate models only on training tasks is warranted. We run evaluation for single rating for the Skywork-Critic-8B and 70B models, two extremely competitive pairwise models. These models are trained exclusively on pairwise tasks, and as such, it is not fair to compare such models against models trained to do single rating. We emphasize that these results are for illustrative purposes only.
>
> We prompt the Skywork critic models using a minimally modified version of the Skywork prompt template to perform single rating evaluation. We preserve the original formatting of the pairwise prompt provided by Skywork, minimally changing the instructions to describe a new task (single rating) and to ask the judge to output its decision as [[an integer between 1 and 5]]. Using this prompt template, both the 8B model is only capable of producing outputs “[[A]]” or “[[B]]”, indicating that training has caused such a model to overfit to the pairwise comparison output format. The 70B model is more capable of producing integer outputs, but such outputs correlated extremely poorly with human ratings (Pearson correlation of effectively 0.00 for Skywork-Critic-70B vs. 0.76 for our 70B model).

---

> ### Author Response · Authors · 2024-11-18
> **Response to Reviewer xfuc (Part 3)**
>
> > In Table 5, what are the 6 types of biases? And How do you measure these biases? I understand the position bias can be measured by the consistency of choosing responses even if the order is swapped. But what about the other biases?
>
> We adopt the biases identified by OffsetBias in their paper [1], and have updated our paper to include a description of the biases and how to interpret such results in Appendix A, starting on line 807.
>
> Table 5 presents results on how immune judge models are to such biases. Each category contains pairwise comparisons, where one response is correct and the other response is incorrect, but in a manner that may “trick” the judge model based on the relevant bias. In other words, if the judge model is susceptible to a particular bias, then it will pick the incorrect response. To measure how immune judges are to these biases, we can simply measure the accuracy of a judge in responding to these pairwise comparisons, where the more unbiased a judge is, the more correct responses they will pick.
>
> To summarize, the six biases that were identified by OffsetBias are as follows:
> - Length bias: judges prefer longer responses, even if the response does not follow the user’s instructions.
> - Concreteness bias: judges prefer responses that are more concrete, such as citing precise percentages or figures, even if they are wrong or irrelevant
> - Empty reference bias: Sometimes the input instruction provided by a user is incomplete (OffsetBias authors provide an example of a user requesting a summary of an article, but forgetting to provide an article). Weaker models are susceptible to hallucinating responses based on imagined input content, whereas strong models ask for clarification. Judges tend to prefer hallucinated model responses rather than responses that ask for clarification.
> - Content continuation bias: judges prefer responses that continue generating related content to user requests, rather than those that faithfully execute user instructions.
> - Nested instruction bias: If the user instruction includes an input (e.g., an article) that includes an instruction, then the judge may evaluate responses based on how well they satisfy the nested response rather than the original user instruction.
> - Familiar knowledge bias: Judge models may prefer responses that contain common information (e.g., idiomatic sayings or common facts) rather than responses that precisely follow the user’s instructions.
>
> The authors of OffsetBias arrived at these six bias categories by analyzing failure modes of judge models by generating many judge responses. Then, the authors perform filtering and manual editing to arrive at samples that fit each of the bias categories.
>
> [1] OffsetBias: Leveraging Debiased Data for Tuning Evaluators, https://arxiv.org/abs/2407.06551
>
> ---
>
> > Line 464 claims that “we plot the average performance across all three evaluation tasks when removing each training task”, but Fig 4 may miss the SFT+DPO w/o CoT critique.
>
> Our objective is to train a generative judge model that produces both judgements AND critiques. As such, removing the training task that includes CoT critique would mean our judge is not trained on preference samples with critiques. Because such critiques can be extremely useful in improving downstream model performance (see the experiment presented in Section 5.5, with results in Figure 6), we choose not to omit this task during our ablations.

---

> ### Author Response · Authors · 2024-11-22
> **Gentle reminder for Reviewer xfuc**
>
> Dear reviewer,
>
> This is a gentle reminder that the discussion period ends on Wednesday, November 26. We are checking in to see if you have any questions or concerns about our response or our revised paper, which was re-written to address reviewer questions and includes new experiments exploring critique quality (Appendix C.7) and training sample difficulty (Appendix C.6).
>
> If you believe our rebuttal and updates address your concerns, it would be greatly appreciated if you would consider increasing your score. We are also more than happy to provide additional clarification and answer any follow-up questions.
>
> Thank you!

---

> > ### Author Response · Authors · 2024-11-25
> > **Follow-up reminder for Reviewer xfuc**
> >
> > Dear reviewer,
> >
> > This is a follow-up reminder that the discussion period is ending soon. The last day for reviewer posts is tomorrow (November 26) and the last day for us to respond is Wednesday (November 27). We want to ensure we have enough time to address any potential follow-up questions or concerns you may have with our response and/or our updated paper.
> >
> > Thanks again!

---

> > > ### Author Response · Authors · 2024-11-30
> > > **A gentle reminder**
> > >
> > > A gentle reminder: if the reviewer has any further concerns, we would be happy to address them.
> > >
> > > Thanks!

---

### Official Review · Reviewer_MyLG · 2024-11-09

**Soundness:** 3
**Presentation:** 3
**Contribution:** 2
**Rating:** 3
**Confidence:** 3

**Summary:**

The paper "Direct Judgement Preference Optimization" improves large language models (LLMs) used as generative judges by introducing Direct Preference Optimization (DPO), which enhances evaluation by learning from both positive and negative examples. The authors train models on pairwise comparison, single ratings, and classification tasks using a mix of human and synthetic data, including Chain-of-Thought (CoT) critiques and response deduction. Their models outperform baselines like GPT-4o across 13 benchmarks, mitigate biases, and adapt to various evaluation protocols, offering more robust and flexible performance while also providing valuable feedback for improving other models.

**Strengths:**

**Diverse Evaluation Tasks**: The paper proposes a comprehensive evaluation framework, including pairwise comparisons, single ratings, and binary classification. This makes the model adaptable to different evaluation needs​.

**Model Performance**: The paper demonstrates that their judge models outperform existing baselines, such as GPT-4o and specialized judge models, across multiple benchmarks, including pairwise comparison, single rating, and classification tasks​.

**Bias Mitigation**: The model robustly counters common biases, such as position and length bias, offering improved generalization across evaluation protocols.

**Weaknesses:**

**Content organization **: The organization of this paper could be further improved, for example, structurize some long paragraphs.


**Design philosophy**: The paper could be strengthened by providing a more detailed explanation of the design philosophy, including the rationale for including each component, their individual functions and relationships, and how they relate to the background presented in Section 2. An ablation study or discussion of the impact of removing individual components would also be valuable.

**Questions:**

See the Weaknesses.


The authors could enhance the paper by clarifying the relationship between Direct Judgement Preference Optimization and Direct Preference Optimization, highlighting key similarities and differences. Additionally, they could more explicitly state the main takeaways of the paper and explain how their use of three types of data for DPO differs from and improves upon the original DPO approach.

What are the takeaway messages of this paper? It seems like the authors use three types of data for DPO rather one type in the original DPO?

---

> ### Author Response · Authors · 2024-11-18
> **Response to Reviewer MyLG (part 1)**
>
> > The organization of this paper could be further improved, for example, structurize some long paragraphs.
>
> Thank you for the feedback regarding the organization of the paper. We have taken the following steps to improve this aspect of the paper. Please see the updated paper for changes; all major additions or changes to the text have been written in blue font.
> - We have broken up longer paragraphs (for example, related work in Sec. 6) and have edited sections for conciseness.
> - We have re-arranged the order of figures and tables such that tables appear closer to relevant text.
> - We have re-written portions of the introduction to clarify our contributions and other relevant work. For example, see lines 73-75 and 82-87.
>
> ---
>
> > The paper could be strengthened by providing a more detailed explanation of the design philosophy, including the rationale for including each component, their individual functions and relationships, and how they relate to the background presented in Section 2. An ablation study or discussion of the impact of removing individual components would also be valuable.
>
> Thank you for your insightful feedback regarding the design philosophy of our work!  Below, we highlight our philosophy for utilizing paired preference data in training our judge model, describe the specific reason for including each of the three DPO training tasks, and recap the ablation study presented in the main body. For each point, we provide references to the main text where such explanations appear. If the reviewer has specific questions about design choices, we are happy to further clarify during the discussion period!
>
> 1. The use of both positive and negative training examples:  The core of our idea is to overcome the limitations of SFT, which only trains the judge model with positive examples, by training with paired preference data. We discuss the limitations of SFT in Section 2 (Lines 146-152) before presenting our approach of utilizing DPO to train our model with three types of preference data in Section 3.
>
> 2. Three types of preference data: The rationale of each type of preference data is summarized in the first paragraph of Section 3. Specifically, we explain that (1) the CoT critique is for boosting reasoning while providing interpretability, (2) the standard judgment is for direct supervision and (3) response deduction is for enhancing reasoning. Each following subsection elaborates these rationales before introducing the corresponding method. For example, Section 3.2 explains why the training signal from CoT critique alone is insufficient and how the standard judgement can address this gap. We have also included Figure 3 to visually support our claim.
>
> 3. Ablation Study: Indeed we have conducted the ablation study in Section 5.3, where we systematically remove one certain component from each baseline to validate our design choices, as acknowledged by Reviewers xfuc and azFf. The results shown in Figure 4 demonstrate that our complete method yields the best overall performance, underscoring the importance of each component.
>
> If the reviewer has further questions, we are happy to clarify!

---

> > ### Author Response · Authors · 2024-11-18
> > **Response to Reviewer MyLG (part 2)**
> >
> > > The authors could enhance the paper by clarifying the relationship between Direct Judgement Preference Optimization and Direct Preference Optimization, highlighting key similarities and differences. Additionally, they could more explicitly state the main takeaways of the paper and explain how their use of three types of data for DPO differs from and improves upon the original DPO approach.
> >
> > > What are the takeaway messages of this paper? It seems like the authors use three types of data for DPO rather one type in the original DPO?
> >
> > Thank you for your valuable suggestions! Below, we outline the relationship between our work and DPO and summarize the key takeaways. We have tried to make these contributions explicit in our updated paper in lines 59-75 and 82-87.
> >
> > **Clarification of our work vs. DPO**: Direct Judgment Preference Optimization is an advanced framework that leverages the foundational principle of DPO which is to learn from both positive and negative examples while extending its applicability to auto-evaluation tasks. Unlike traditional DPO, which relies on a singular type of preference data, our work integrates multiple types of training data--CoT critique, standard judgment and response deduction. This integration enriches the learning process by providing a more comprehensive training signal, thereby enhancing our model's evaluation capability.
> >
> > **Key takeaways:**
> >  - **Learning from both positive and negative examples is essential for improving the evaluation capability of LLM judges.** Existing works primarily train their judge models on positive examples, which is insufficient due to the limitation of SFT. To mitigate this, we train our judge with both positive and negative examples, enabling them to discern between good and bad judgments. This approach significantly enhances the evaluation capabilities of the judges, as evidenced by our experimental results on an extensive evaluation suite of 13 benchmarks, which we present in Section 5.1.
> >  - **Diverse preference data types enhance judgment preference learning:** Our advanced preference learning framework leverages three distinct types of preference data, each contributing uniquely to the evaluation capabilities of an LLM judge:
> >    - **Chain-of-Thought (CoT) Critique:** This enhances the judge's reasoning skill, which is crucial for improving performance and providing interpretability. As shown in Section 5.5, the CoT critique can also serve as natural language feedback to enhance downstream models.
> >    - **Standard Judgment:** This provides direct supervision, aligning the judge more closely with human labels.
> >    - **Response Deduction Preference:** This type focuses on deepening the understanding of the characteristics of good and bad responses.
> > - Our comprehensive experiments verify the effectiveness of our method (Section 5.1). Notably, we demonstrate that a more capable judge can more robustly address biases (Section 5.2), adapt flexibly to user-specified evaluation instruction (Section 5.4) and even provide AI feedback for model development (Section 5.5).

---

> > > ### Comment · Reviewer_MyLG · 2024-12-02
> > > **Thanks for the response.**
> > >
> > > The paper could be further improved by:
> > >
> > > - Explain the functions of ``Chain-of-Thought Critique, Standard Judgement, and Response Deduction``, and highlight their necessity. Clarify the relationship between these three components.
> > > - What are the benefits of combining data from ``Single Rating, Pairwise Comparison, and Classification``? You may include case studies or quantitative analysis. In which scenarios does using only type of data fail, but combining another type proves effective?   Using more data seems an advantage over DPO, but what is real postive consequence? which challenges does it address by using diverse data? Does it help DPO  generize well？
> > > - The writing could be further improved by better organizing the content, for example: structurize lengthy  content and make them logically coherent.

---

> > > > ### Author Response · Authors · 2024-12-02
> > > > **Response to Reviewer MyLG**
> > > >
> > > > > Explain the functions of Chain-of-Thought Critique, Standard Judgement, and Response Deduction, and highlight their necessity. Clarify the relationship between these three components.
> > > >
> > > > It would be helpful if the reviewer could give us an example of what they feel is unclear about the functions of these three types of data in our paper. We are happy to further clarify and revise our paper to reflect such clarifications.
> > > >
> > > > We have already summarized the rationale in our initial response and have written at length about these types of data in our paper in Section 3. This section outlines what CoT critique, standard judgement, and response deduction data consist of, and our rationale for including each of these types of data. Additionally, as we mentioned in our initial response, we have conducted an ablation study, which other reviewers have noted, on these three types of DPO data.
> > > >
> > > > Does the reviewer have concrete questions about the content of Section 3?
> > > >
> > > > > What are the benefits of combining data from Single Rating, Pairwise Comparison, and Classification?
> > > >
> > > > We emphasize that we are training specialized LLM-as-judge models with DPO. Single rating, pairwise comparison, and classification specific evaluation capabilities of LLM-as-judge models. That is, the goal of our paper (and a line of prior work) is to train an LLM to perform pairwise comparison evaluation, single rating evaluation, and binary classification evaluation. These three capabilities are explained in Section 2. These are common evaluation capabilities, and all existing prior work train models to perform at least a subset of these capabilities (See Table 1 in the paper for concrete examples).
> > > >
> > > > > You may include case studies or quantitative analysis. In which scenarios does using only type of data fail, but combining another type proves effective?
> > > >
> > > > We emphasize to the reviewer that these are specific capabilities we train our LLM-as-judge to have, and as such, we include all three in our training data mix. Training models to perform multiple evaluation capabilities has been shown in prior work to be mutually beneficial. Examples below; we will also update the paper to include this rationale.
> > > >
> > > > “Unlike other datasets designed for pairwise preference task, Ultrafeedback and Helpsteer datasets are designed for single scoring task. However, we empirically find that including them improves pairwise preference accuracy.” [1], Section 5.1
> > > >
> > > > “...multitask instruction tuning effectively equips the model with general-purpose quality assessment capabilities.” [2], Section 4.1
> > > >
> > > > “...training on pairwise ranking improves direct assessment performance” [3], Section 6.3
> > > >
> > > > [1] OffsetBias: Leveraging Debiased Data for Tuning Evaluators, https://arxiv.org/abs/2407.06551
> > > >
> > > > [2] Foundational Autoraters: Taming Large Language Models for Better Automatic Evaluation, https://arxiv.org/abs/2407.10817v1
> > > >
> > > > [3] Prometheus 2: An Open Source Language Model Specialized in Evaluating Other Language Models, https://arxiv.org/abs/2405.01535
> > > >
> > > > > Using more data seems an advantage over DPO, but what is real postive consequence? which challenges does it address by using diverse data? Does it help DPO generize well？
> > > >
> > > > We believe that our experimental results indicate that our approach helps judge models generalize across a variety of evaluation settings (pairwise comparison, single rating, classification) and types of evaluation (e.g., evaluating model response safety with HHH or summarization capabilities with InstruSum). This point is highlighted in our paper, lines 80 - 81: “This holistic approach allows our models to better generalize to various evaluation tasks, as we demonstrate with our experiments.”
> > > >
> > > > Concretely, our training approach has yielded a family of judge models with the best aggregate performance across 13 benchmarks, outperforming models trained to perform only one of these evaluation tasks. Please see Section 4 for a description of our evaluation set-up, and section 5 for experimental results. As a concrete example of generalization, our 70B model, trained to perform all 3 of the evaluations, outperforms Skywork-Critic-70B, a model trained *only to perform pairwise comparisons, in terms of average performance on pairwise comparison benchmarks!* This result holds for other pairwise-specific and single-rating specific models, indicating better generalization on evaluation tasks.
> > > >
> > > >
> > > > > The writing could be further improved by better organizing the content, for example structurize lengthy content and make them logically coherent.
> > > >
> > > > Can the reviewer point us to exact lines, pages, or paragraphs where they feel the writing is incoherent or overly long? We have already re-written and re-arranged portions of the paper for readability (see our initial comment for exact changes). However, we are happy to continue revising the paper based on this discussion period!

---

> ### Author Response · Authors · 2024-11-22
> **Gentle reminder for Reviewer MyLG**
>
> Dear reviewer,
>
> This is a gentle reminder that the discussion period ends on Wednesday, November 26. We are checking in to see if you have any questions or concerns about our response or our revised paper, which was re-written to address reviewer questions and includes new experiments exploring critique quality (Appendix C.7) and training sample difficulty (Appendix C.6).
>
> If you believe our rebuttal and updates address your concerns, it would be greatly appreciated if you would consider increasing your score. We are also more than happy to provide additional clarification and answer any follow-up questions.
>
> Thank you!

---

> > ### Author Response · Authors · 2024-11-25
> > **Follow-up reminder for Reviewer MyLG**
> >
> > Dear reviewer,
> >
> > This is a follow-up reminder that the discussion period is ending soon. The last day for reviewer posts is tomorrow (November 26) and the last day for us to respond is Wednesday (November 27). We want to ensure we have enough time to address any potential follow-up questions or concerns you may have with our response and/or our updated paper.
> >
> > Thanks again!

---

> > > ### Author Response · Authors · 2024-11-30
> > > **Follow-up**
> > >
> > > A gentle reminder: if the reviewer has any further concerns, we would be happy to address them.
> > >
> > > Thanks!

---

### Author Response · Authors · 2024-11-18
**Thank you to all reviewers!**

We thank the reviewers for their helpful comments and suggestions. Overall, we are delighted that reviewers found our work “well-motivated” and “novel” (Reviewer azFf). Reviewers found “The novelty [...] in the way of constructing preference data” (Reviewer xfuc) and our auxiliary training task of response deduction as a “nice idea” (Reviewer azFf). We are also delighted that all reviewers found our evaluations to be “comprehensive” (Reviewer MyLG) and “extensive” (Reviewer xfuc), with our results being “convincingly better compared to our SOTA models” (Reviewer azFf) and “demonstrating superiority over existing baselines” (Reviewer Ay6r). Reviewers also found our judge showed “good reliability” in the face of bias (Reviewer Ay6r) and found the additional analysis with prompt flexibility, downstream model improvement, and training task ablations as “all nice additions in the paper” (Reviewer axFf).

We have updated our manuscript with additional experimental results showing the effects of hard negative preference training pairs (Appendix C.6) and evaluating the quality of critiques generated from our models (Appendix C.7). We have also re-written portions of the main text for clarity in response to reviewer questions. All major additions or revisions are marked in blue text.

We respond to each reviewer’s comments separately, and hope to have a productive discussion period!

---

### Meta-Review · Area_Chair_EHtm · 2024-12-22

**Metareview:**

This paper proposed to apply DPO to train a model for evaluation. This works collected and constructed and mixes preference data from multiple sources and trained evaluation models. Experiments demonstrated the effectiveness of the proposed method.

Strengths:
1. Strong empirical results across various tasks.

Weaknesses:
1. As one reviewer pointed out, this work mainly applies DPO for training an evaluation model, and the contribution mainly lies in constructing preference data and empirically training the judge models, and similar evaluation data has been collected by previous works such as Vu et al., 2024, Want et al, 2024c, and criticGPT.

Overall, despite the strong empirical results, this work mainly applies DPO for training judge models, and the main contribution is mainly in constructing preference data. Considering the review scores, I'm recommending rejection, but I wouldn't mind if the paper gets accepted.

**Additional Comments On Reviewer Discussion:**

A major concern from reviewers is that compared existing literature the contribution seems incremental. The authors clarified that they trained the judge model to also generate explanations, which is novel. However, this work still seems to be an application of DPO to judge model training at a high level.

---

### Decision · Program_Chairs · 2025-01-22

Reject